# FSVM: A Few-Shot Threat Detection Method for X-ray Security Images

**DOI:** 10.3390/s23084069

**Published:** 2023-04-18

**Authors:** Cheng Fang, Jiayue Liu, Ping Han, Mingrui Chen, Dayu Liao

**Affiliations:** Tianjin Key Lab for Advanced Signal Processing, Civil Aviation University of China, Tianjin 300000, China; cfang@cauc.edu.cn (C.F.);

**Keywords:** X-ray images, baggage threat detection, few-shot learning, support vector machine

## Abstract

In recent years, automatic detection of threats in X-ray baggage has become important in security inspection. However, the training of threat detectors often requires extensive, well-annotated images, which are hard to procure, especially for rare contraband items. In this paper, a few-shot SVM-constraint threat detection model, named FSVM is proposed, which aims at detecting unseen contraband items with only a small number of labeled samples. Rather than simply finetuning the original model, FSVM embeds a derivable SVM layer to back-propagate the supervised decision information into the former layers. A combined loss function utilizing SVM loss is also created as the additional constraint. We have evaluated FSVM on the public security baggage dataset SIXray, performing experiments on 10-shot and 30-shot samples under three class divisions. Experimental results show that compared with four common few-shot detection models, FSVM has the highest performance and is more suitable for complex distributed datasets (e.g., X-ray parcels).

## 1. Introduction

X-ray baggage screening is a crucial component of security inspection. With the rise of deep learning, many automatic threat detection methods based on deep-learning methods have been proposed to increase the effectiveness of security screening and lower the labor intensity of human operators [1]. Deep neural networks, which benefit from deeper layer depth and a larger hypothesis space than conventional opitimization algorithms, can be trained with large amounts of data to achieve expressive results [2]. However, automated detection methods based on supervised learning are highly dependent on the quantity and reliability of annotations in datasets. For X-ray security datasets, collecting large amounts of well-labeled images for each type of contraband is extremely difficult due to the wide variety of contraband categories. Threats in the same categories also have different morphologies. Current public datasets for X-ray baggage only support a few contraband categories, which is far from sufficient to meet practical needs [3]. Furthermore, the high demand for annotations will make the threat detection model more difficult to respond to and employ, increasing the security risk. It is also common for criminals to carry unconventionally shaped contraband, such as shaped knives and weapons, to avoid security inspection, which is more likely to be overlooked. There is a great need to construct threat detection models when annotated samples are scarce.

Big data models are difficult to achieve satisfactory results from a limited number of samples. For recognition tasks, a better embedding space enables features to be more separable on the hyperplane [4,5]. However, models, in the data-hunger setting, may end up with an overfitted representation of the problem. In recent years, few-shot learning(FSL) defines it as an optimization problem in a limited data regime [6]. To acquire a discriminative feature, a few-shot model must utilize some kind of prior knowledge, e.g., reducing parameters to avoid overfitting or searching for a good initialization of the hypothesis space [7,8]. Various few-shot approaches have been proposed, which have promoted the progress of this research field. However, many of them focus on natural image datasets, and there is a lack of research in other subdivided fields. For now, few-shot learning methods have been applied to aerial images [9,10] and SAR images [11,12]. There are also works that focus on chest X-ray images for disease diagnosis [13,14]. However, few-shot researches for X-ray security datasets and for threat detection are still under-studied. For X-ray security datasets, prohibited items in baggage are usually cluttered or occluded, especially in the reality scene. It cause X-ray security datasets have a more complex distribution than natural datasets [15]. Figure 1 shows image samples of X-ray imaging baggage. The cluttered nature of X-ray baggage also negatively affects the performance of few-shot detection methods. There is a great need to specifically address this issue and provide solutions for X-ray security images.

To bridge this gap, this work proposed a few-shot threat detection method for X-ray security images. Given the complex distribution of X-ray security contraband, we constructed an SVM module to embed it into the original base detector for a few-shot learning task. Within the SVM embedding module, an additional supervised classification task was built to transfer the higher-order decision information into the former layers. The higher-order information, which is in the form of gradients, will back-propagate to fine-tune the unfrozen parameters. Through the end-to-end training process, the proposed method FSVM will produce the embedding space more suitable for complex distributed data. Experimental evidence proves that FSVM can support situations where the number of threat samples is extremely scarce and achieve better results compared with other few-shot object detection methods.

Our contributions can be concluded as follows:We proposed a few-shot threat detection model called FSVM for X-ray security images, enabling the model to detect novel contraband items with only a small number of samples.To generate an informative embedding space, an SVM embedding module is proposed, which can be trained end-to-end, and embedded it freely into an object detection model.The SVM loss is proposed as a part of the joint loss function. This additional constraint, used in the fine-tuning stage, will back-propagate the supervised information to former layers. Trained with this joint loss, the proposed model is more suitable for few-shot application scenarios.

## 2. Related Work

### 2.1. Automatic Threat Detection for X-ray Security Images

The X-ray imaging principle is that the X-ray tube generates beams that penetrate the scanned items. Depending on the material density, high-density items and low-density items on the X-ray images have distinguishable colors, which facilitates the manual screening process. However, prohibited items are occluded from each other in the single-view imaging, giving different forms to the same type of contraband. The cluttered nature of baggage makes the task harder. Furthermore, the human operators need sufficient knowledge and experience to achieve a confident inspection, which is hard to cultivate. Operators in practice could not maintain an average performance under excessive work intensity.

To tackle the above problems, automated detection methods have been proposed in recent years. With the development of deep learning methods, deep neural networks are showing their superb learning and generalization abilities. Akcay et al. [16] applied a CNN-based transfer learning method on the X-ray baggage security dataset, achieving 88.5% mean average precision(mAP) on a six-class object detection task and 97.4% mAP on a two-class firearm detection task. Hassan et al. [17] enhanced ROI features by extracting contour information from the same object placed at different angles, and detection on SIXray and GDXray achieved 96.8% mAP, exceeding the human-recognizable accuracy of 94.5%.

Supervised methods have shown promising performance in X-ray threat detection, while the lack of annotations limits contemporary deep model training. In response to this weakness, Xu et al. [18] proposed a method for identifying and utilizing attention mechanisms to generate heatmaps intermediately to localize prohibited items. However, this approach still requires sufficient labels to specify the category information.

### 2.2. Few-Shot Learning

Data collection has become one of the bottlenecks in the progress of deep learning. A child, however, could learn novel concepts by reviewing a small amount of data and annotations. To overcome the learning gap between human and artificial intelligence, few-shot learning (FSL) has become a promising research area. It aims at improving the model effect with only a few annotated samples (usually under 30-shots) while getting as close as possible to many-shot models. Initially, few-shot learning only focused on few-shot classification (FSC) [6], but many few-shot object detection (FSOD) methods have emerged in recent years for practical applications. FSOD is more complex than classification as it requires not only classifying categories but also predicting location boxes [19].

In few-shot learning tasks, categories are divided into base/seen classes and novel/unseen classes. Base classes have sufficient annotated data, and novel classes only have a few. Several transfer learning based FSOD methods first train on base classes, and fine-tune on novel classes [20,21]. Wang et al. [20] suggests a transfer learning method, which can effectively improve the model’s generalization of the novel class while alleviating the model’s forgetfulness of the base class. For few-shot learning tasks, a representative embedding space can enable a representative feature for novel class samples [22]. Sun et al. [21] combines a contrast learning approach to guide parameter updates by constructing CPE losses. The model maps the proposed features to a feature space with smaller intra-class distances and larger inter-class distances, allowing similar instances of different classes to be more distinguishable. Our method conducts research based on such methods.

### 2.3. End-to-End Trainable Embedding Layer

Enriching the representational information extracted by deep learning networks is key to improving the recognition ability of the model [23], and this representational information is expressed as a high-dimensional vector. Metric learning is used to determine whether two high-dimensional vectors are in the same class by calculating the distance between them in an embedding space [24].

To obtain a more transferable and representative embedding space, a range of approaches are proposed, which prove that the end-to-end training can exploit the search process of deep architectures in the hypothesis space to explore the invariant properties behind the data [25,26,27]. Some of them consider second-order statistics, such as covariance operators and gaussian operators, as an end-to-end trainable embedding layer to constrain the model parameter updating [25,26]. Such methods have now been widely used in unsupervised domain adaptation tasks, where they have been shown to yield a more informative embedding representation and more complex distribution [28,29]. In order to fuse higher-order information into few-shot object detection models, the literature [27] gives us the idea of embedding the quadratic programming solution process into FSOD, which is able to back-propagate higher-order information by solving the optimal solution of the convex function with the help of the features that constitute the covariance matrix.

## 3. Method

### 3.1. Problem Formulation

For dataset D={(x,y),x∈X,y∈Y}, each sample (x,y) consist of the input image *x* and associated annotation $y\in \{\left(c_{i},l_{i}\right),i=1,2,\cdots ,N\}\}$, which denotes its categories and bounding box coordinates 1 of N object instances in the image *x*. Categories are divided into two subsets, the base class $C_{base}$ and the novel class $C_{novel}$, respectively. $C_{base}\cap C_{novel}=\emptyset$. For each contraband instance, $c_{i}\in \left\{C_{n}\cup C_{b}\right\}$. In a few-shot learning scenario, the N-way K-shot setting means an N-class classification task, where each category has K samples. With respect to the dataset division, $D_{base}$ have many examples and $D_{novel}$ only have K examples for each category.

The few-shot training comprises of two stages: the base-training stage and the fine-tuning stage. In the base-training stage, the training dataset consists of all base-class data $D_{base}$, and the detection model is trained to obtain sufficient semantic information common to X-ray images. In the fine-tuning stage, we only use a small number of sample to verify the performance of the few-shot model. For each category $c_{i}$ in *C*, *K* samples are taken from both novel class and base class to form the $D_{finetune}$, with a sample size of C×K. In fine-tuning training process, model randomly sampling data from $D_{finetune}$ and adjusting the parameters by gradient descent algorithm. Knowledge learned from the frist stage will transfer to novel classes during fine-tuning stage. The model validates its performance in both the base class and the novel class in the test set.

### 3.2. The SVM-Constrained FSOD Architecture

Figure 2 shows the overall architecture of our model. A two-stage detection model, Faster R-CNN, is used as the base detector. Backbone network and FPN receive the input images and output feature maps as a proportionally sized feature maps at multiple levels, in a fully convolutional fashion. The Region proposal network (RPN) proposes candidate boxes from the feature map. RoI pooling extracts the proposed region corresponding to the anchor frame and unifies it into a fix-sized feature vector. The produced region proposal $P=\{\vec{p}\in R^{m},y\}$, where $\vec{p}$ is the m-dimension RoI feature vector, will be calculate in the R-CNN predictor head. Then, RoI head output box offsets and class confidences. In fine-tuning stage, the amount of data is far less than base-training stage, so the overfitting problem will be very severe. To avoid the overfitting, we freeze the parameters of the backbone network and fine-tune the remaining parameters. The derivable SVM embedding module participate in the few-shot fine-tuning training. This embedding layer adds SVM multiclassification task as a constraint on the fine-tuning training phase. The additional supervised information will enrich the representation of the embedding space. Details about the SVM module are described in Section 3.3.

### 3.3. Svm Embedding Module

As a powerful and robust algorithm, support vector machines (SVMs) play an important role in traditional classification and regression tasks [30]. The mechanism of SVM is to achieve a hyperplane that could maximize the interval between classes by solving a convex optimization problem, which has advantages especially for low-quantity data regimes. In this part, we utilized the characteristics of the SVM algorithm as an additional constraint to guide the optimization direction for few-shot tasks. Instead of using SVM as a normal classifier, we embedded our proposed SVM module into the base detector, which could be trained end-to-end and therefore enrich the representation of our model.

The SVM module we have designed is shown in Figure 3. The region features $\vec{p}$ are remapped with $f_{z}\left(x\right):R^{m}\to R^{d}$, before being transported to the SVM module. The projected features ${\vec{p}}^{\prime }=f_{z}\left(\vec{p}\right)$ will pass through an IoU filter. It will filter out a portion of the proposed regions under an IoU threshold. Experimentally, setting the IoU threshold can effectively control the quality of the features that input to the SVM layer, resulting in a better performance of the model. The above filtered features will input into the SVM module. Inspired by the literature [27], the multi-classification SVM solution algorithm [31] is placed into our proposed SVM module. The objective function of the multiclassification SVM algorithm is:(1)$$\begin{array}{c} f_{svm}=\underset{\left\{\omega_{c}\right\}\left\{\xi_{i}\right\}}{\arg\min}\;\sum_{c}^{C}\left|\right|\omega_{c}{\left|\right|}_{2}^{2}+\beta \sum_{i}^{n}\xi_{i}\\ s.t.\;\;\omega_{y_{i}}P^{\prime }-\omega_{c}P^{\prime }+\delta_{y_{i},c}⩾1-\xi_{i} \end{array}$$
where $P^{\prime }\in R^{n\times d}$ represent the feature matrix which consist of former feature vector ${\vec{p}}^{\prime }$, β is the regularization parameter and $\delta_{y_{i},c}$ is the Kronecker delta function. We convert Equation (Equation 1) into the dual formulation, which can reduce the dependence on the embedding dimension. Let
(2)$$\omega_{c}\left(\alpha^{c}\right)=\sum \alpha_{n}^{c}{\vec{p}}_{n}^{\prime }$$

After adding the dual set of variables, we can get the Lagrangian of the dual program as stated in [31],
(3)$$\max-\sum_{i,j}\left({\vec{p}}_{i}^{\prime }\cdot {\vec{p}}_{j}^{\prime }\right)\left(\alpha_{i}^{c}\cdot \alpha_{j}^{c}\right)+\beta \sum_{i}\alpha_{i}^{y}\cdot {\bar{1}}^{y_{i}}\;s.t.\;\;\forall i\;\;\alpha_{i}^{y}⩽{\bar{1}}^{y_{i}}\;\;\mathrm{and}\;\;\sum_{c}\alpha_{i}^{c}\cdot \bar{1}=0$$
where ${\bar{1}}^{y_{i}}$ represents the one-hot encoding labeled vector, $\bar{1}$ is the vector whose components are all one. $\alpha^{c}$ is our required solution, where $\alpha^{c}\in R^{n\times c}$. Let $\theta ={\left\{\alpha^{c}\right\}}_{c=1}^{C}$ be the parameters computed in the dual function, we abstract the above equation into the following optimization problem:(4)$$\begin{array}{cc} minimize\;\; & f_{0}\left(\theta ,{\vec{p}}^{\prime }\right)\\ subject\;to\;\; & f(\theta ,{\vec{p}}^{\prime })⪯0\\ \; & h(\theta ,{\vec{p}}^{\prime })=0 \end{array}$$
where $f_{0}\left(\theta ,{\vec{p}}^{\prime }\right)$ is the convex function, $f(\theta ,{\vec{p}}^{\prime })$ and $h(\theta ,{\vec{p}}^{\prime })$ are the inequality constraint and equation constraint of this optimization problem, respectively, where θ is the parameter of the SVM embedding layer. According to Equations (1) and (2), $\theta =f_{svm}\left({\vec{p}}^{\prime }\right)$, where ${\vec{p}}^{\prime }$ is the proposed region feature, which has ${\vec{p}}^{\prime }=f_{z}\left({\vec{p}}^{\prime }\right)$. In order to make the SVM layer trainable end-to-end, it is necessary to obtain $\frac{\partial f_{\theta }}{\partial f_{p}}$ as gradient information to back-propagate, which can also be written as $\frac{\partial f_{svm}}{\partial f_{z}}$(Figure 3). Assume that ${\vec{p}}^{\prime }$ in Equation (Equation 4) satisfy $S\left({\vec{p}}^{\prime }\right)=\left\{\theta |g\left(\theta ,{\vec{p}}^{\prime }\right)=0\right\}$, where $g:R^{d}\times R^{c}\to R^{c}$ is an implicit function (implicit function) about θ and ${\vec{p}}^{\prime }$. For the function $f:R^{n}\to R$, the gradient information is denoted as $\nabla_{x}f\left(x\right)\in R^{n}$. For the function $f:R^{n}\to R^{m}$, the gradient information is denoted as Jacobian matrix $D_{x}f\left(x\right)\in R^{m\times n}$. Then the gradient $\frac{\partial f_{\theta }}{\partial f_{p}}$ is the Jacobian matrix $D_{p}S\left({\vec{p}}^{\prime }\right)\in R^{d\times c}$, which is the gradient we need to obtain.

We can establish the Lagrangian equation according to Equation (Equation 4):(5)$$L\left(\theta ,\lambda ,v,{\vec{p}}^{\prime }\right)=f_{0}\left(\theta ,{\vec{p}}^{\prime }\right)+\lambda^{T}f\left(\theta ,{\vec{p}}^{\prime }\right)+v^{T}h\left(\theta ,{\vec{p}}^{\prime }\right)$$

Since the SVM objective function in the dual formulation in Equation (Equation 3) satisfies the strong duality of Slater’s condition, and both the equation constraint and the inequality constraint are differentiable, the gradient can be solved [31]. This KKT condition for the Lagrangian function can be written in the form of an implicit function.
(6)$$g\left(\theta ,\lambda ,v,{\vec{p}}^{\prime }\right)=\left[\begin{array}{c} {▿}_{\theta }L\left(\theta ,\lambda ,v,{\vec{p}}^{\prime }\right)\\ diag\left(\lambda \right)f\left(\theta ,{\vec{p}}^{\prime }\right)\\ h(\theta ,p) \end{array}\right]$$

According to Barratt [32], when partial derivatives in $L(\theta ,\lambda ,v,{\vec{p}}^{\prime })$ all exist, by the implicit function theorem, set $g(\theta ,\lambda ,v,{\vec{p}}^{\prime })=0$, we can obtain:(7)$$D_{p}S\left({\vec{p}}^{\prime }\right)=-D_{\theta }g{\left(\tilde{\theta },\tilde{\lambda },\tilde{v},{\vec{p}}^{\prime }\right)}^{-1}D_{p}g\left(\tilde{\theta },\tilde{\lambda },\tilde{v},{\vec{p}}^{\prime }\right)$$
where both $D_{\theta }g\left(\tilde{\theta },\tilde{\lambda },\tilde{v},{\vec{p}}^{\prime }\right)$ and $D_{p}g\left(\tilde{\theta },\tilde{\lambda },\tilde{v},{\vec{p}}^{\prime }\right)$ can be solved. Equation (Equation 7) can be used to back-propagate the gradient. The QP solver suggested by Amos and Kolter [33] completes this part of the computation. Since the whole computational environment is in the tensor environment for deep learning, we can obtain the gradient from the SVM layer to the remapping layer by this Jacobian matrix.

We can incoporate kernel functions to prove its convergence. The objective function with kernel is as follows.
(8)$$\max-\sum_{i,j}K\left({\vec{p}}_{i}^{\prime }\cdot {\vec{p}}_{j}^{\prime }\right)\left(\alpha_{i}^{c}\cdot \alpha_{j}^{c}\right)+\beta \sum_{i}\alpha_{i}^{y}\cdot {\bar{1}}^{y_{i}}\;s.t.\;\;\forall i\;\;\alpha_{i}^{y}⩽{\bar{1}}^{y_{i}}\;\;and\;\;\sum_{c}\alpha_{i}^{c}\cdot \bar{1}=0$$
where $K({\vec{p}}_{i}^{\prime }\cdot {\vec{p}}_{j}^{\prime })$ is the kernel function of the feature, and different kernel functions can be chosen to bring into the above equation. In Section 4, we choose the two most common kernel functions, i.e., linear kernel function and Gaussian kernel function, for our experiments, denoted by FSVM-L and FSVM-G, respectively, see Section 4. The expressions of the linear and Gaussian kernel functions are:(9)$$\begin{array}{c} K_{linear}\left({\vec{p}}_{i}^{\prime },{\vec{p}}_{j}^{\prime }\right)={\vec{p}}_{i}^{\prime }\cdot {\vec{p}}_{j}^{\prime } \end{array}$$(10)$$\begin{array}{c} K_{gaussian}\left({\vec{p}}_{i}^{\prime },{\vec{p}}_{j}^{\prime }\right)=\exp\left(-\frac{\left|\right|{\vec{p}}_{i}^{\prime }-{\vec{p}}_{j}^{\prime }{\left|\right|}^{2}}{2\sigma^{2}}\right) \end{array}$$

### 3.4. Joint Loss Funtion

To measure the probability of its classification correctness, we use the negative log-likelihood as the SVM loss, which compares the similarity between the predicted distribution and the one-hot label distribution.
(11)$$L_{svm}=-\sum_{n}1_{y_{n}}\cdot \log\left(\alpha_{c}\cdot {\vec{p}}_{n}^{\prime }\right)$$

In the two-stage training process, the base-training phase uses the standard Faster R-CNN loss [34], namely RPN loss, classification loss and box regression loss. In the fine-tuning stage, adding the SVM loss constraint described above, we end up with a joint loss function of:(12)$$L=L_{rpn}+L_{cls}+L_{bbox}+\gamma L_{svm}$$
where γ is a scale parametor used to adjust the loss balance. The SVM loss will back-propogate the gradients of SVM decision information to former layers successfully.

## 4. Experiment

In this section, we evaluate our approch and compare it with four common few-shot object detection methods, including meta-learning method, transfer learning method, and pretrained transfer learning methods. Results of the former FSOD methods on the X-ray baggage security dataset are provided for the follow-up research, and we provide ablation experiments.

### 4.1. Dataset

The X-ray baggage security inspection image dataset SIXray [35] was collected from subway stations, and there are 8929 manually annotated X-ray baggage images for five different classes: gun, knife, wrench, pliers, and scissors. It comprises of objects with a wide range of scale, viewpoint, and overlap, resulting in heavy noise in dataset. We cleaned and removed a small portion of images in SIXray, which included images with low resolution, noticeable changes in color distribution, and some images that lacked usable labels. The cleaned dataset contains 8827 images in total. Each image may contain multiple contrabands, and there are about 14,000 contraband samples available for training. In this paper, the dataset is partitioned into a training set and a testing set, where the ratio is about 4:1.

### 4.2. Experimental Setups

#### 4.2.1. Category Division

The dataset is randomly divided into three novel splits, each with three base classes and two novel classes. K = 10, 30 is selected for the N-way K-shot setting to test the model’s generalizability under various sample sizes. The novel classes in tree divisions are Split1: gun, knife; Split2: pliers, wrench; Split3: scissors, gun.

#### 4.2.2. Implementation Details

We adopt Faster R-CNN [34] with Resnet-101 and a feature pyramid network (FPN) as a base detector. All experiments were performed on an Nvidia GeForce 3090 GPU using the momentum SGD optimizer with a weight decay of 0.0001. In the base-training stage, we train for 60 epochs with a batch size of 8 and a learning rate of 0.02. To make the solution space more stable, base training starts with a warm-up strategy to adjust the learning rate gradually. The learning rate in the fine-tuning stage was set to 0.01 and the batch size is set to 4. For calculating the SVM loss, we use label smoothing with a smoothing parameter set to 0.1.

#### 4.2.3. Evaluation Metrics

We use the mean average precision (mAP) to describe the performance of detection results. This metric takes into account both the precision and recall of a certain category, by calculating the area under the Precision-Recall Curve. Precision and recall is determined by:(13)$$\begin{array}{c} Precision=\frac{TP}{TP+FP}\\ Recall=\frac{TP}{FP+FN} \end{array}$$
where TP represents the positive samples that are correctly identified, FP represents the positive samples that are incorrectly identified, and FN represents the negative samples that are incorrectly identified. The average precision is obtained by:(14)$$AP=\int_{0}^{1}PdR$$And the mAP take the average of AP values of all categories. In the few-shot setting, the model performance will consider both base class and novel class as follows:(15)$$\begin{array}{cc} mAP= & \frac{1}{C}\sum_{i=0}^{C}AP_{i}\\ bAP= & \frac{1}{C_{base}}\sum_{i=0}^{C_{base}}AP_{i}\\ nAP= & \frac{1}{C_{novel}}\sum_{i=0}^{C_{novel}}AP_{i} \end{array}$$
where *C*, $C_{base}$ and $C_{novel}$ represents the total number of categories, the number of base classes and the number of novel classes, respectively.

### 4.3. Comparison Experiment

As in Table 1, we conduct comparison experiments on four common few-shot object detection techniques: the Meta-RCNN [36], FsdetView [37], TFA [20], and FSCE [21]. As with our proposed model, all methods use ResNet101 as the backbone network and FPN for feature enhancement. Among the four comparison methods, the Meta R-CNN algorithm is a meta-learning method, FsdetView is a transfer learning method, and both TFA and FSCE are pre-trained transfer learning methods. The experiment shows that the methods without using pre-trained models (Meta R-CNN, FsdetView) fail to achieve the performance of pre-trained methods (TFA, FSCE). Results show that our method is still superior compared to the two pre-trained methods and obtains the best results in the novel class for all three different divisions.

In this paper, the before-and-after changes of the bAP are also counted (Table 2). Compared to other methods, FSVM is able to maintain the base class AP at a higher level after completing fine-tuning. Like mentioned in [21], this means the base forgetting is much less after fine-tuning, i.e., the model could learn new information of the novel classes while remaining the characteristics of the base classes.

### 4.4. Ablation Experiment

To determine the details of the designed model and to understand its impact, the ablation experiments are reported. If not specifically indicated, all ablation experiments were done based on Novel split1 10shot SIXray data. To maintain the consistency, same hyper-parameters were used for all experiments.

#### 4.4.1. FSVM

In principle, the SVM embedding layer needs to back-propagate the learned supervised information in the form of gradients through the additional classification task, and the parameter updatting occurs in the fine-tuning phase depend on the quality of the back-propagated gradients, so the following ablation experiments are conducted for the designed SVM module, as shown in Table 3. To achieve a meaningful gradient, the remap layer (consist of two fully connected layers) is used for remapping the feature input. The experimental results indicate that the model with SVM constrained supervision has higher performance both at 10-shot and 30-shot setting. Table 3 also shows that add remapping layer can enhance the effect of SVM supervision, in addition to freezing part of the parameters can also reduce overfitting.

We found that adding a remapping layer can effectively reduce the SVM loss generated by our additional multi-classification task shown as Figure 4. During the first 250 iters of the fine-tuning stage, the remapping layer was able to bring down the SVM loss, while the SVM loss without the remapping layer always oscillated above and below the initial value. Since the parameter space of the remapping layer is smaller than that of the overall model, we analyze that the remapping layer can act as a gradient amplifier by adjusting the effect of SVM layer to amplify a meaningful and expressive SVM losses.

#### 4.4.2. Dimension Control

Experiment shown in Table 4 confirm the mapped feature dimension size. We test the impact of remapping layers from low (16) to high (4096) dimensions. The experiments show that although increasing the dimensionality of the remapping layer can remap the features to a high-dimensional space and make them separable using its sparsity property, using high-dimensional vector mapping when the number of samples is insufficient leads to overfitting, which makes the model accuracy decrease in the high-dimensional (4096) case. We found that nAP50 performs smoothly at the dimensiona equal to 128, which also can obtain more accurate detection boxes (higher nAP75) compared to other fetched values.

#### 4.4.3. Input Feature Control

The quality of the features we input to the SVM layer becomes very crucial for the proformance. We use an IoU filter to control the quality of the input RoI feature. First, we obtained the RoI feature with a detected bounding box, while checking whether the RoI passed the IoU requirement. The greater the overlap between the detected bounding box and the groundtruth, the higher the input quality. Meanwhile, it also means that the number of RoI features to the SVM module will be reduced.

To validate the method, we set the values of IoU requirement to 0.5 and 0.7, respectively, and test them on 10-shot and 30-shot setting, as shown in Table 5. The result shows that the IoU threshold = 0.7 can effectively improve the model performance, proves that a higher IoU threshold in our IoU filter can promote the quaility of RoI features for the SVM multi-classification task, which can generate a better constrained SVM gradient. Further more, the improvement is more obvious when the number of sample size is smaller.

#### 4.4.4. The Embedded SVM Layer Control

In this paper, we use two most common kernel functions for testing, namely linear kernel functions and gaussian kernel functions. The linear kernel is usually applied when the original feature vector is already relatively linearly separable, while the gaussian kernel can construct nonlinearly separable decision intervals by mapping the original feature vector to a gaussian kernel space. We test the impact of these two kernel functions on the model, shown in Table 6. The performance of the two kernel functions is measured by changing the regularization parameter β and the balance factor of SVM loss γ. The regularization parameter β can regulate the solution of the multi-classification SVM task in Equation (Equation 1), and a smaller β leads to simpler constitutive decision space. It can also reduce the overfitting in the data-hunger setting. We tested two values of β, 0.1 and 0.5, respectively. Table 6 shows that a larger β leads to a higher performance, β=0.5 is more suitable for the current sampled sample points. However, fine-tuning the balance fator γ produces a slight perturbation of the model parameters and no significant enhancement effect. So we further test the value of γ while setting β to 0.5 in two kernel-type scenarios. Figure 5 shows the experiment results in both 10-shot and 30-shot settings. Results show that the model achieves the best results in the 10-shot and 30-shot scenarios when lambda is equal to 0.5. For the gaussian kernel-based FSVM, nAP50 will decline when lambda is greater than 0.5. For the linear kernel-based FSVM, nAP50 will decline when lambda is greater than 0.8. Finally, we determined γ=0.5 to maintain model stability.

### 4.5. Visual Results and Comparison

The detection result are provided in Figure 6. The trained detector was fine-tuned by Novel Split1 with 30-shot samples. Each contraband is denoted in the figure by a bounding box and a category with a confidence score. The novel classes in Novel Split 1 are knife and gun. Results show that the detector can successfully detect novel contrabands after a few steps of finetuning under only 30-shot samples.

Figure 7 compares the results of our FSVM method to two transfer-based few-shot methods, TFA and FSCE.We show the visualization results for three different scenarios: (a) clear background with no shading or overlap; (b) laptop-obscured; and (c) complicated background with items obscured and cluttered. As shown in the picture, although the predicted bounding box is not accurate enough, our model detects contraband in all three scenarios and all targets in scenario c).

We also use Gradcam [38] to visualize the region of interest for the model when making classification decisions. Figure 8 shows the Gradcam heatmap results for the knife class. The upper part of the figure represents original X-ray baggage samples in the dataset. We can see that our model mostly recognizes a knife by its handle when the items are occluded by other metal structures. The activated regions (the red part of the picture) on samples are compact and concentrated, which shows our model learned the correct representation message through SVM-module finetuning. To further compare the Gradcam results, we also visualized the heatmap results of TFA and FSCE as shown in Figure 9. In this figure, we focus on gun class, and we can see that both TFA and FSCE focus not only on the prohibited items but also more on the complex background. The result shows that our model mitigates the negative impact of contraband that is difficult to distinguish in complex background distributions to some extent, which reflects the high learning ability of the model.

## 5. Conclusions

In this paper, we propose a few-shot object detection method, FSVM, which is designed for complex distributed X-ray security images with contraband. The FSVM can detect threats in X-ray baggage with an extremely insufficient number of samples. To increase detection performance, a end-to-end trainable SVM module is used to embed it into the object detection network. This allows for the back-propogation of additional supervised information to create a better embedding space for few-shot tasks. Four commonly used FSOD methods are tested to incorporate the few-shot learning method into X-ray threat detection, including the meta-learning method, the transfer learning method, and pre-trained transfer learning method. Results for the 10 shot and 30 shot few-shot samples are presented on the X-ray baggage security dataset SIXray. Compared with the four FSOD methods, Table 1 shows that our method not only has highest average precision in both 10-shot and 30-shot settings, but also alleviates base forgetting. However, the confusion matrix results of our model showed more leak detection in novel classes than in base classes. Though our proposed model achieves better results, it is hard to avoid leak detection in the few-shot setting, especially in the cluttered and occluded X-ray security dataset. In the future, we will focus on promoting the quality of proposals produced by RPN and further improving the detection results.

## Figures and Tables

**Figure 1 sensors-23-04069-f001:**
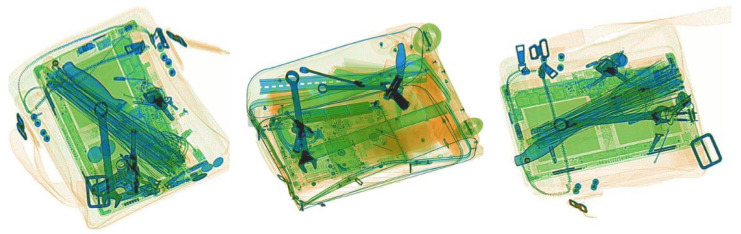
Baggage samples in the X-ray security dataset. As shown in the pictures above, prohibited items (Knife, Gun, Wrench) will be cluttered together, making their outline complicated. They are also easily occluded by the compact structures, e.g., laptops and metal zippers.

**Figure 2 sensors-23-04069-f002:**
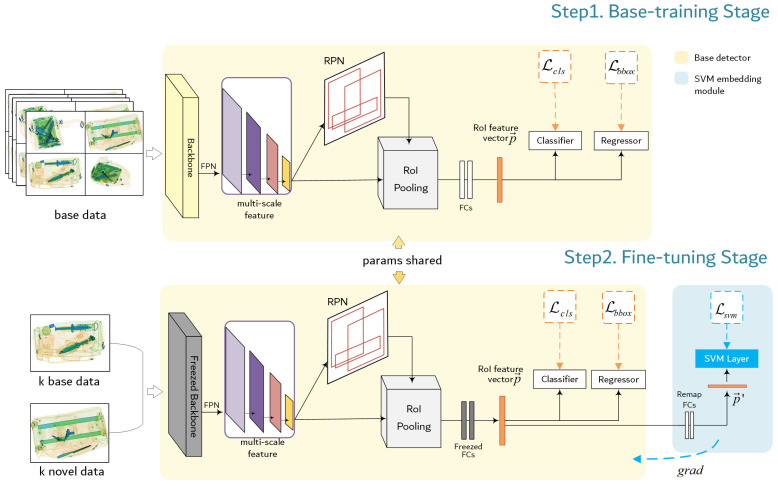
The SVM-constrained few-shot threat detection model. The yellow background notes the base detector, Faster R-CNN. In the fine-tuning stage, our proposed SVM module (signed with a blue background) finetune the unfrozen parameters in the yellow part and does not participate in the final model inference.

**Figure 3 sensors-23-04069-f003:**
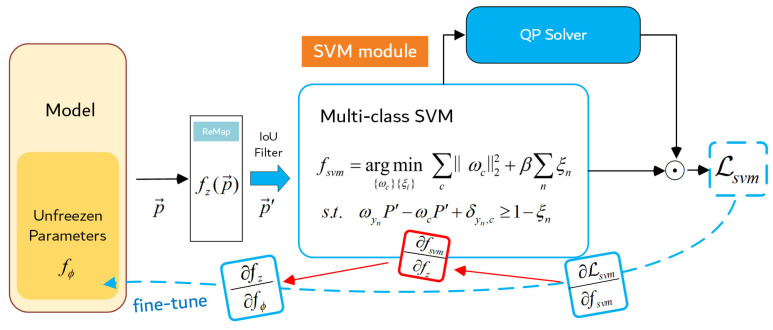
SVM module. Our proposed SVM module consists of a remap layer, an IoU filter, the multi-class SVM object function, and the QP Sovler. The SVM module is trained end-to-end on the deep neural network for additional supervised decision-making. Gradients calculated from $L_{svm}$ will back-propogate to the former layers relying on the chain rule.

**Figure 4 sensors-23-04069-f004:**
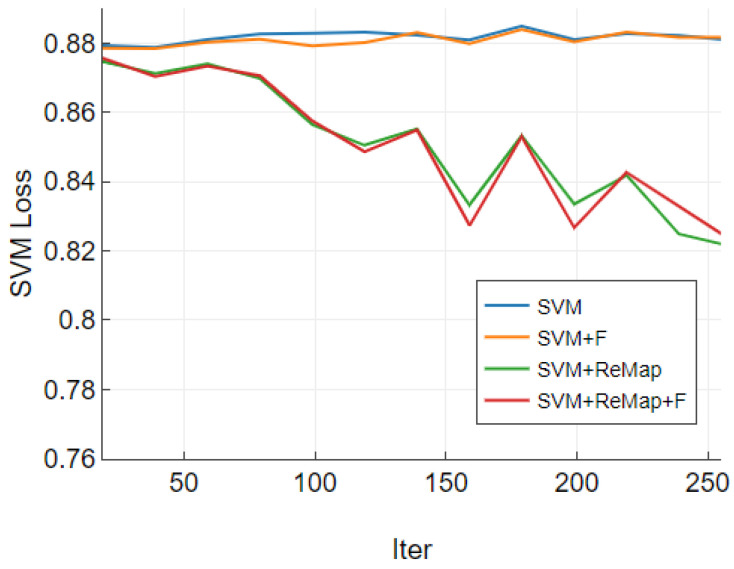
SVM loss after using remapping layers.

**Figure 5 sensors-23-04069-f005:**
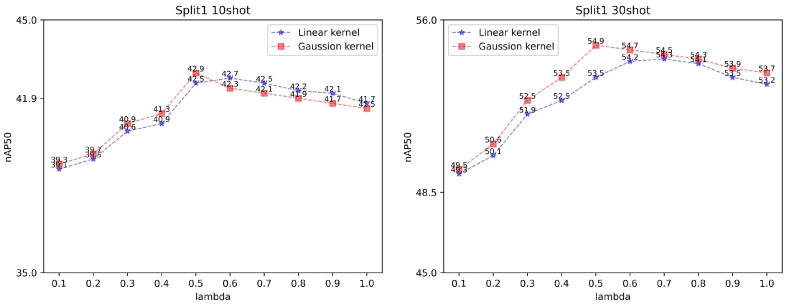
Determination of γ. We test the nAP50 values for γ from 0.1 to 1.0.

**Figure 6 sensors-23-04069-f006:**
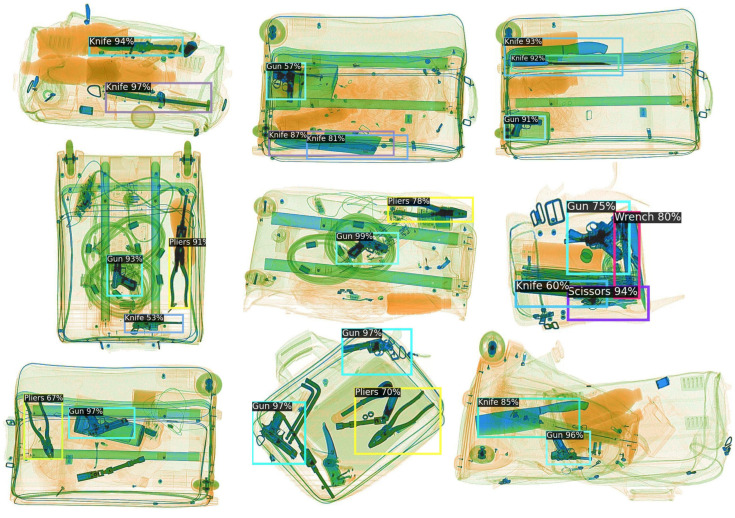
Detection results under the Novel Split1 setting. The novel classes in Novel split1 are gun and knife. Our model can effectively detect the novel class object in cluttered baggages by learning from only 30-shot instances.

**Figure 7 sensors-23-04069-f007:**
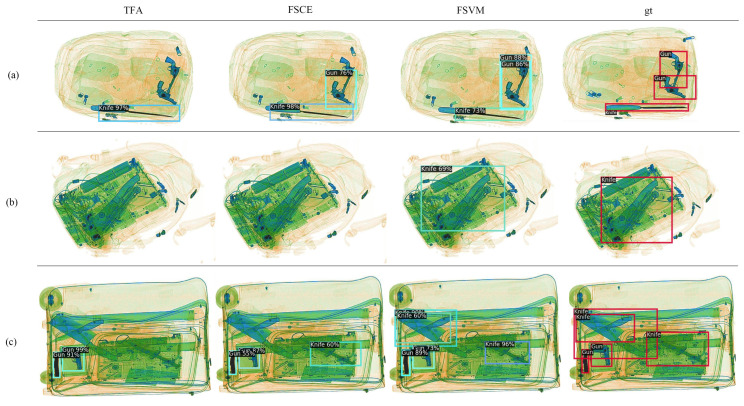
Detection results comparison. we select three different scene to show the adventage of our model. (**a**) clear background, no shading and overlap, (**b**) occluded by an laptop and (**c**) complicated background, and items are occluded and cluttered.

**Figure 8 sensors-23-04069-f008:**
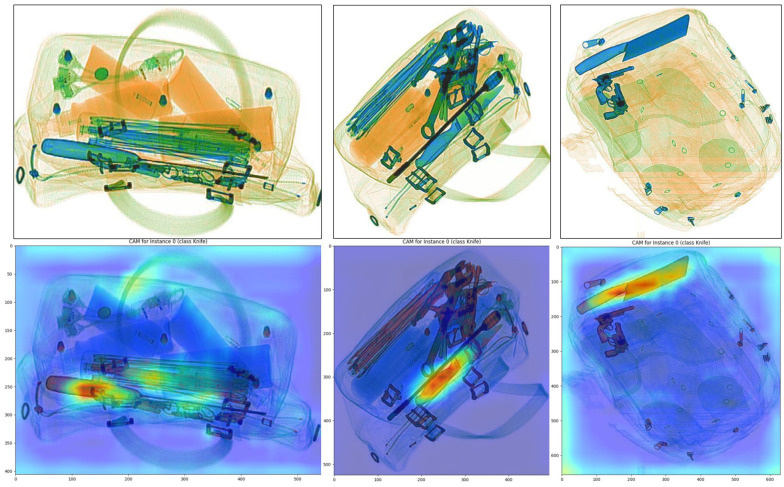
GradCam heatmap results of our model. Figure shows the activated level of every region in each samples for the category of knife. The activation levels from low to high are shown with blue to red.

**Figure 9 sensors-23-04069-f009:**
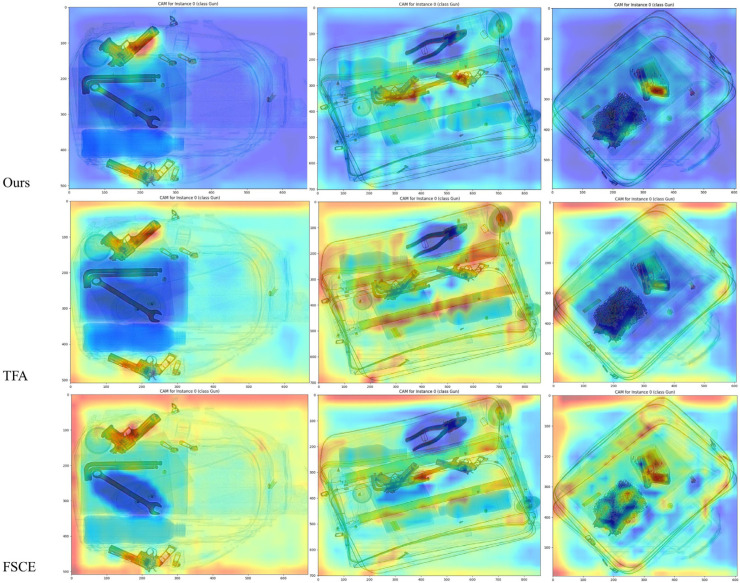
GradCam heatmap results comparison for the category of Gun. The activation levels from low to high are shown with blue to red.

**Table 1 sensors-23-04069-t001:** Performance evaluation of three SIXray category divisions. For each novel split, the results are reported by bAP and nAP. FSVM-L and FSVM-G are SVM-constrained methods with the linear kernel and the gaussian kernel, respectively. * represents results from 10 random seeds. We set the best results in red and the second-best results in blue.

(a) 10-Shot Result												
	Novel Split1				Novel Split2				Novel Split3			
Model	bAP50	Gun	Knife	nAP50	bAP50	Pliers	Wrench	nAP50	bAP50	Scissors	Gun	nAP50
Meta R-CNN [36]	76.8	0.8	10.7	5.8	80.3	12.4	4.5	8.4	73.9	0.9	9.1	5.0
FsdetView [37]	74.7	1.1	11.5	6.3	77.7	11.3	12.8	12.1	76.9	1.3	5.5	3.3
TFA [20]	78.5	31.3	15.0	23.2	78.1	11.5	9.4	10.5	77.1	15.5	37.3	21.4
FSCE [21]	79.8	50.9	24.4	37.7	80.3	24.9	10.2	17.5	82.2	24.3	50.3	37.3
FSVM-L * (Ours)	83.4	52.4	28.9	40.6	83.4	26.6	10.1	18.3	85.8	31.8	52.9	42.2
FSVM-G * (Ours)	85.7	54.8	26.6	40.7	82.8	26.8	12.5	19.6	84.5	33.1	55.4	44.2
**(b) 30-Shot Result**												
	**Novel Split1**				**Novel Split2**				**Novel Split3**			
**Model**	**bAP50**	**Gun**	**Knife**	**nAP50**	**bAP50**	**Pliers**	**Wrench**	**nAP50**	**bAP50**	**Scissors**	**Gun**	**nAP50**
Meta R-CNN [36]	75.9	0.4	10.4	5.4	79.9	9.3	4.6	6.9	76.2	3.5	9.1	6.30
FsdetView [37]	78.6	9.1	11.6	10.3	78.3	14.5	12.2	13.4	76.4	5.8	5.2	5.50
TFA [20]	80.9	39.8	14.1	26.9	83.0	15.4	16.5	15.9	78.8	14.7	39.5	27.1
FSCE [21]	85.4	62.8	29.8	46.2	85.7	25.6	16.5	21.1	82.5	36.4	63.0	49.7
FSVM-L * (Ours)	88.6	70.1	34.6	52.4	87.2	31.1	19.4	25.2	86.7	40.9	65.7	53.3
FSVM-G * (Ours)	88.4	69.3	38.9	54.1	88.4	35.2	23.1	29.1	85.1	41.7	66.8	54.2

**Table 2 sensors-23-04069-t002:** Base forgetting comparison. Since only base data are input during base-training stage, the closer the bAP value is to mAP after fine-tuning, the more resistant the model is to the base-class forgetting.

	Base-Training	Fine-Tuning	
Model	mAP50	bAP50	nAP50
TFA [20]	82.9	80.9	26.9
FSCE [21]	90.8	85.4	46.2
FSVM-L (Ours)	90.8	**88.6**	52.4
FSVM-G (Ours)	- -	88.4	**54.1**

**Table 3 sensors-23-04069-t003:** Ablation experiments of SVM embedding layer design details. nAP are used to measure the performance of the model.

SVM Module	Remap Layer	Freeze FCs.	Novel mAP50	
			10-Shot	30-Shot
** **	** **	** **	39.6	49.4
√	** **	** **	42.0	52.1
√	** **	√	42.5	53.6
√	√	** **	43.4	53.9
√	√	√	**44.9**	**54.5**

**Table 4 sensors-23-04069-t004:** Choice of the remapped dimension. The model performance is tested by nAP value when the IoU of the detector is 50 and 75.

Dim =	16	64	128	256	1024	4096
nAP50	44.7	45.5	45.6	**45.7**	45.4	43.9
nAP75	6.9	7.2	**7.9**	7.5	7.3	7.4

**Table 5 sensors-23-04069-t005:** Performance with different IoU filter threshold.

IoU	10 Shot		30 Shot	
	nAP50	nAP75	nAP50	nAP75
0.5	39.7	3.8	**51.6**	7.4
0.7	**41.7**	**5.4**	**51.6**	**8.7**

**Table 6 sensors-23-04069-t006:** Parameter adjustment of the SVM module. nAP50 are tested by varying the regularization parameter β and the loss balance factor λ. We used 30-shot Novel split1 data, and fixed random seeds to keep the experiment consistent.

	Linear Kernel		Gaussian Kernel	
β	**0.1**	**0.5**	**0.1**	**0.5**
γ=0.5	52.7	53.5	53.8	54.9
γ=0.7	52.2	54.3	53.4	54.5

## Data Availability

Due to the nature of this research, participants of this study did not agree for their data to be shared publicly, so supporting data is not available.

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
