# Peer review of "FSVM: A Few-Shot Threat Detection Method for X-ray Security Images"

_sensors, 2023, doi:10.3390/s23084069_

Round 1

Reviewer 1 Report

The paper proposes a few-shot SVM-constraint threat detection model named FSVM for detecting rare contraband items in X-ray baggage. The paper highlights the challenge of acquiring well-annotated images, especially for rare contraband items. FSVM aims to detect unseen contraband items with only a few labeled samples.

 The paper proposes a novel approach by embedding a derivable SVM layer to back-propagate the supervised decision information into the former layers, instead of simply finetuning the original model. The paper also creates a combined loss function utilizing SVM loss as an additional constraint.

 The authors evaluate FSVM on the public security baggage dataset SIXray, performing experiments on 10-shot and 30-shot samples under three class divisions. The experimental results show that FSVM outperforms four common few-shot detection models and is more suitable for complex distributed datasets like X-ray parcels.

Below are some comments that authors can address for the final version:

1. Cite more references for few shot method applied to other applications. The good place is line 38.

2. What is the additional constraint created by the combined loss function utilizing SVM loss in FSVM? Elaborate this in Methods section.

3. What is the main limitation of the proposed method, and what will be the focus of future work?

4. Please provide the confusion matrix for the final tuned method applied to the test dataset.

Thank you!

Reviewer 2 Report

This manuscript is well-organized and easy to follow. The authors studied a FSVM model for automatic threat detection from x-ray security images.  The experimental results are promising when comparing with other common methods.

I only have a minor comment: In the joint loss function of Eq.(12), are the first three losses of similar range? If not, why were scale parameters not included before them? Also, how was the parameter \lambda determined in the experiments?
